# Pathophysiology of Sepsis and Genesis of Septic Shock: The Critical Role of Mesenchymal Stem Cells (MSCs)

**DOI:** 10.3390/ijms23169274

**Published:** 2022-08-17

**Authors:** Matthieu Daniel, Yosra Bedoui, Damien Vagner, Loïc Raffray, Franck Ah-Pine, Bérénice Doray, Philippe Gasque

**Affiliations:** 1Unité de Recherche en Pharmaco-Immunologie (UR-EPI), Université et CHU de La Réunion, 97400 Saint-Denis, France; 2Service de Médecine d’Urgences-SAMU-SMUR, CHU de La Réunion, 97400 Saint-Denis, France; 3Laboratoire d’Immunologie Clinique et Expérimentale Océan Indien LICE-OI, Université de La Réunion, 97400 Saint-Denis, France; 4Service de Médecine Interne, CHU de La Réunion, 97400 Saint-Denis, France; 5Service d’Anatomo-Pathologie, CHU de la Réunion, 97400 Saint Pierre, France; 6Service de Génétique, CHU de La Réunion, 97400 Saint-Denis, France

**Keywords:** mesenchymal stem cells, circulating MSCs, pericytes, perivascular MSCs, innate immunity, inflammation, sepsis, septic shock, immunomodulation, miRNA, exosomes

## Abstract

The treatment of sepsis and septic shock remains a major public health issue due to the associated morbidity and mortality. Despite an improvement in the understanding of the physiological and pathological mechanisms underlying its genesis and a growing number of studies exploring an even higher range of targeted therapies, no significant clinical progress has emerged in the past decade. In this context, mesenchymal stem cells (MSCs) appear more and more as an attractive approach for cell therapy both in experimental and clinical models. Pre-clinical data suggest a cornerstone role of these cells and their secretome in the control of the host immune response. Host-derived factors released from infected cells (i.e., alarmins, HMGB1, ATP, DNA) as well as pathogen-associated molecular patterns (e.g., LPS, peptidoglycans) can activate MSCs located in the parenchyma and around vessels to upregulate the expression of cytokines/chemokines and growth factors that influence, respectively, immune cell recruitment and stem cell mobilization. However, the way in which MSCs exert their beneficial effects in terms of survival and control of inflammation in septic states remains unclear. This review presents the interactions identified between MSCs and mediators of immunity and tissue repair in sepsis. We also propose paradigms related to the plausible roles of MSCs in the process of sepsis and septic shock. Finally, we offer a presentation of experimental and clinical studies and open the way to innovative avenues of research involving MSCs from a prognostic, diagnostic, and therapeutic point of view in sepsis.

## 1. Concept of Sepsis and Genesis of Septic Shock

### 1.1. Definitions

Sepsis and septic shock represent the consecrated terms to designate the systemic response, in terms of severity, generated by the innate immune system (IIS) in the presence of a pathogen. This aggression, in connection with the infection, is responsible for a sequence of molecular and cellular interactions that triggers an inflammatory response. This host response may be dysregulated, inducing an immune system dysfunction. This dysregulated host response associated with at least one organ dysfunction, identified as an acute change ≥ 2 in Sepsis-Related Organ Failure Assessment score (SOFA score) currently defines the “sepsis” according to SEPSIS-3 criteria [1]. In most cases, this inflammation is controlled and balanced by the host immune system and the disease is only represented by a reaction called “infection” often accompanied by a systemic inflammatory response syndrome formerly named “SIRS” [2]. On the other side, if the triggering is too massive (pathogen’s virulence) or associated with an old age, underlying comorbidities, concurrent injuries, medications or genetic predispositions, the inflammatory phase is racing to a more severe form: the “sepsis” which can often lead to the “septic shock”. Sepsis and, *a fortiori*, septic shock are responsible for the onset of acute organ dysfunctions or even a multiple organ dysfunction syndrome (MODS) requiring the management of patients in intensive care units (ICU). MODS associated with sepsis represents one of the main causes of morbidity and mortality worldwide in ICU with more than 50% of in-hospital deaths linked to sepsis [1,3]. It is commonly accepted that, when accompanied by MODS, septic shock reflects the exacerbated response to the presence of the pathogen and frequently leads to immune, metabolic, and hematological dysfunctions [4,5]. The different definitions related to sepsis and their continuum from infection to the picture of multiple organ failure are presented in Figure 1.

### 1.2. Physiopathology and Treatments of the Septic Shock

#### 1.2.1. Cell-Mediated Innate and Adaptative Immune Responses

The triggering event of a septic state is represented by the colonization of an organ, compartment or fluid of the organism by a pathogen inducing a local inflammatory reaction. Resident macrophages and dendritic cells, known as professional antigen presenting cells (APCs), are the first line of defense [6]. These cells are able to recognize pathogen-associated molecular patterns (PAMPs) using specific pathogen recognition receptors (PRR), including Toll-like receptors (TLRs). The first elements involved in the genesis of this IIS reaction are the PAMPs but also the host-derived debris released by damaged cells and known as damage (or danger)-associated molecular patterns (DAMPs). These specialized components interact with PRRs as TLRs, C-type lectin receptors (CTLRs), NOD-like receptors (NLRs), and RIG-I-like receptors (RLRs) [7]. DAMPs such as defensins, cathelicidin (LL-37), eosinophil-derived neurotoxin, and High Mobility Group Box-1 (HMGB-1) are also able to activate these PRRs and are called alarmins [8]. Activation of PPRs results in the engagement of complex metabolic cascades responsible for the formation of the inflammasome [9]. The inflammasome is, in turn, responsible for the synthesis and release of pro-inflammatory cytokines among which we can cite, interleukin-1-beta (IL-1ß), tumor necrosis factor-alpha (TNF-α), interleukin-6 (IL-6), interleukin-18 (IL-18), and HMGB-1 [10] sometimes carrying out the cytokine released syndrome or “cytokine storm” during an exacerbated activity of the inflammasome. The activation of the PRRs and the cytokine release are also responsible for a cascade of reactions leading, in immune cells, to a nuclear translocation of the nuclear factor kappa B (NFκB) [11] and induction of the transcription of targets genes involved in the synthesis of other pro-inflammatory cytokines including IL-1ß, IL-6, and chemokines such as IL-8 (CXCL8) [12]. This inflammatory microenvironment is essential to provoke the release of molecules such as complement proteins involved in direct lysis of the pathogen and to promote the recruitment of effector cells of the innate and then the adaptive immune system (AIS). In turn, these effector cells recruited at the site of the attack, will participate in the management and destruction of the pathogen.

The adaptive immune system (AIS) response is time-lagged relative to IIS. It consists of presenting the pathogen incriminated by the APC to the naive T lymphocytes (LyT) of the nearby secondary lymphoid organs (SLO). These LyT will, for the most part, migrate to the site of infection to participate in cell-mediated immunity. A portion of these T lymphocytes remains in SLO to elicit humoral mediated immunity by activating B lymphocytes (LyB) specific for the presented antigenic motifs [13]. There are currently several types of adaptive immune responses depending on signaling and differentiation pathways [14]. Among these different ways we can cite the T helper 17 (Th17) pathway, a pathway defined by differentiation into T helper 1 (Th1) lymphocytes promoting cell-mediated immunity and a T helper 2 (Th2) pathway for humoral-mediated immunity. This differentiation is determined by the cytokine pattern secreted in response to infection. The Th1 response is induced by interferon-gamma (IFN-γ) secreted by Th1 cells and by IL-12 produced by APCs. It is accompanied by the activation of cytotoxic CD8 + lymphocytes and monocytes. The Th2 response is triggered by the secretion of IL-4 and by the expression of CD40 ligand on the surface of macrophages and LyT and allows the activation of LyB secreting immunoglobulins directed against the constituents of microorganisms and/or their toxins [15].

#### 1.2.2. Peripheral Neuro-Immune Control of the Inflammatory Process in Sepsis

All the initial inflammatory reactions normally resolve once the pathogen has been controlled by the organism, giving way to anti-inflammatory processes to form the basis for tissue repair. Patients with a form of infection with a so-called “adapted” immune response are characterized by the appearance of anti-inflammatory factors allowing to counterbalance the inflammatory response after the initial phase which, when not controlled, leads to a deleterious chronic inflammatory syndrome. The early neuroinflammatory modulation results notably in an activation of the autonomic nervous system (ANS) with a reflex loop involving the descending parasympathetic (vagus nerve) and sympathetic (orthosympathetic system by ganglia nodosis) pathways [16,17,18]. Tracey et al. thus showed that activation of the vagus nerve by signals originating from the IIS in response to the presence of DAMPs enables neuromodulation of the immune response [18]. This reflex loop carries out a relay at the level of the brainstem and leads to the activation of acetylcholine (Ach) expressing LyT cells (ChAT+ cells), representing a particular lymphocyte population expressing the choline acetyltransferase (ChAT) enzyme and playing a role in inhibiting TNF-α levels [19,20], regulating blood pressure [21] as well as circulating cytokine levels in mouse [19]. Activation of the ANS stimulates the secretion of Ach and norepinephrine in the spleen, resulting in an activation of the population of ChAT+ and, among other effects, an inhibition of the macrophages producing pro-inflammatory cytokines via an Ach-dependent alpha7-nicotinic acetylcholine receptor activation [19]. This biofeedback allows curbing the inflammatory response and prevents a “runaway” of the immune response. Several studies have highlighted a defect in the regulatory processes of the immune response in severe sepsis patients [22] and to PAMPs and DAMPs [23].

There are also important interactions between the ANS, notably the sympathetic system and innate or adaptive immunity through specific receptors [24]. Adrenergic receptors thus constitute receptors belonging to the ANS, classified into different categories and having a role in modulating the immune response. These receptors are mainly located in the primary and secondary lymphoid organs and are responsible for down-regulation of the immune response [25]. Among IIS cells, neutrophils (polymorphonuclear neutrophils, PNN) strongly express some adrenergic type receptors, particularly beta-2 receptors [26]. Their activation is responsible for a decrease in the production of reactive oxygen species (ROS), the migration of PNNs, and the formation of neutrophil extracellular traps (NETs) involved in bactericidal activities [26,27].

#### 1.2.3. The Balance between Pro-Inflammatory and Anti-Inflammatory Processes

The infectious process is actually described as a balanced state associating concomitant inflammatory (e.g., IL1β, TNF-α) and anti-inflammatory responses (mediated chiefly by IL-10 and TGF-beta2). Sepsis and septic shock states can be interpreted as an imbalance between those two parts of the immune and inflammatory manifestations in favor of an unsuitable reaction of the IIS and AIS. This inappropriate immune response can persist for several hours or days after the initial infectious trigger and can be responsible, at least in part, for damage to various organs. The initial response observed during sepsis is similar to that observed during systemic inflammation of other origins and processes related to ischemia-reperfusion [28]. This immune response dysfunction will have various consequences on the organ functions at the initial site of the infection but also systemically. The genesis of septic shock is accompanied by attacks of multiple systems (neurons, endothelium among others) leading to acute kidney injury (AKI), acute liver injury, myocardial dysfunction, acute lung injury (ALI), and/or acute respiratory distress syndrome (ARDS). Added together and exceeding the host’s adaptive capacities, these dysfunctions lead to the development of multiple system organ failure.

#### 1.2.4. Endothelial Dysfunction and Hemostasis Troubles

The circulating cytokines will activate the endothelial cells which represent one of the first targets to endothelial and cardiovascular dysfunctions [29]. Cardiovascular system failure affects more than 50% of patients with sepsis within 24 h of admission to ICU [30] and is found in this same proportion post-mortem [31]. The vascular endothelium plays a major role in the homeostasis of many richly vascularized tissues such as the cardiovascular, respiratory, cerebral, and renal systems. While septic shock is generally accompanied by an array of MODS, endothelial dysfunctions appear to be the major factor associated with a poor prognosis of patients [32,33]. The endothelium participates in maintaining the internal environment’s homeostasis and regulating the function of the various cardiovascular, pulmonary, renal, and cerebral systems [34,35]. There is a decrease or even a loss of this regulation with the loss of the endothelial barrier integrity [36]. Endothelial dysfunction is a well-described entity now occurring in critical illness. The consequences of this endothelial dysfunction are multiple: loss of regulation of regional microcirculations, ischemia/reperfusion and oxidative stress mechanisms, hemostasis disorders and formation of microthrombi and tissular hypoxia [36]. These disorders result in a deregulation of the balance between cellular energy needs (respiratory cycle) and oxygen and nutrient inputs as well as activation of pro-apoptotic mechanisms [37]. The mechanisms associated with sepsis leading to cell death are still the subject of many studies but the physiopathology of sepsis-associated cell death seems to be multifactorial and linked to both mitochondrial dysfunction (loss of membrane permeability), transcription and oxidative stress with increased cellular ROS production [38,39].

The uncontrolled systemic inflammatory cascade is also responsible for an overactivation of coagulation that can progress to disseminated intravascular coagulation (DIC) with microthrombi formation within the organs, aggravating hypoperfusion lesions. These coagulation disorders result from the activation of pro-coagulant molecules such as thrombin and tissue factor by activated endothelial and monocyte cells and the inactivation of molecules responsible for the fibrinolysis (protein C and endothelial pathways of anticoagulation) [40].

#### 1.2.5. Central Nervous System Involvement in Sepsis

The central nervous system (CNS) is also a major actor in septic states genesis, affected by the aberrant immune response to infection through activation of the ANS (as aforementioned), the hypothalamic pituitary axis, and the release of hormones (corticoids), cytokines and neurotransmitters [41]. The CNS is involved both in detecting the pathogen, cytokines, hormones, and other molecules secreted by immune effectors [42]. The CNS is a part of a general neuroimmune reflex loop and exerts, with the help of the vagus nerve, a negative feedback on the inflammasome by reducing the circulating levels of TNF-α [16] and by activating ChAT+ cells. Interestingly, ChAT+ cells are also known to be derived, at least in vitro, from human umbilical cord mesenchymal stem cells (hUC-MSCs) [43].

This reflex loop also involves the brainstem and in particular the nucleus of the solitary tract, which acts as a nucleus serving as a relay in the higher centers between the afferents and the efferents of the ANS. During septic states, the secretion of proinflammatory cytokines is thus the cause but also the consequence of cerebral aggression and septic encephalopathy by altering the blood-brain barrier (BBB) and activating glial cells [44,45]. 

#### 1.2.6. Sepsis and Immunoparalysis

Sepsis is characterized by a phenomenon called “immunoparalysis” which can be described as an impairment of innate and adaptive immunity responsible for a long-term susceptibility to infections including viral reactivations [46,47]. This affection is accompanied by a decreased survival in septic patients [47]. Septic states particularly impact LyT populations, both from a quantitative and qualitative point of view. Within the LyT population, a distinction is made in particular between CD8^+^ (CD8^+^) defined by a CD56^−^, CD57^−^, CD8^+^ phenotype, natural killer (NK) and natural killer-“type T” cells (NKT) respectively possessing CD56^+^ or CD57^+^ phenotypes [48]. CD8^+^ requiring activation by CD4^+^ T cells and APC, will convey cytotoxic functions attributed to perforin and granzymes directed against infected cells. NK and NKT are cells of innate immunity possessing intrinsic cytotoxicity functions [49]. Among T cells repertory, a decrease in the pool of naive CD8^+^ cells (i.e., antigen-inexperienced T cells) significantly in patients with septic shock as well as an alteration of their cytotoxicity functions has been reported [50]. Furthermore, expression of inhibitory receptors (2B4 and PD-1 receptors) has been reported on the surface of naive post-sepsis T cells, which may increase susceptibility to infection and morbidity and mortality in the host [51,52].

### 1.3. Therapeutics for the Treatment of Septic Shock

Even today, the treatment of septic shock remains the subject of many debates and studies. The current validated treatments, reaffirmed by the Surviving Sepsis Campaign [53], can be divided into two treatment groups. Early treatment (<6 h) consists of the administration of empiric antibiotic therapy and treatment necessary to optimize the flow rates of the macrocirculation and the microcirculation and therefore, to manage the imbalance between inputs and metabolic needs. Maintenance and control, in the first 24 h, of these various biological and clinical parameters improve survival. While the implementation of these recommendations was gradually accompanied by an improvement in patient outcomes through standardization of practices, they were ineffective in obtaining an additional reduction in the overall mortality of these patients. Consequently, many teams are working to evaluate new therapies for sepsis, such as TNF-α antagonist [54], recombinant human activated protein C (APC) [55], intravenous immunoglobulin G therapy [56], TLR4 antagonist [57], IL-1 receptor antagonist [58], and talactoferrin [59].

Neither of these new therapies nor the early goal-directed therapy has shown significant efficacy in the treatment of sepsis in terms of morbidity and mortality [60]. Among the therapies cited, only subgroup analyses were able to show efficacy in certain categories of patients with particularly high mortality [61], leading to the following question: would the development of novel immunoregulatory avenues for instance using mesenchymal stem cells (MSCs) be of clinical benefit in sepsis and septic shock?

The sepsis and a fortiori the septic shock remain pathologies encountered in emergency and intensive care medicine with significant morbidity and mortality [62]. The therapies offered to patients remain ineffective due to the complexity of the pathophysiology of this affection involving, among others, the role of the effectors of innate and adaptive immunity. MSCs, by virtue of their particularly immunomodulatory characteristics (as described below), can represent both major players in the intrinsic control of septic shock and a source for the establishment of new therapeutic alternatives to what currently remains an issue in critical care medicine of the 21st century. The main objectives of this review are to identify the roles played by MSCs during septic states in the detection of the infection to instruct immunity, to understand their implications in the process of uncontrolled sepsis leading to vasoplegic and cardiogenic shock states and their prophylactic and curative therapeutic applications.

## 2. Origins and Roles of Mesenchymal Stem/Stromal Cells

### 2.1. Ontogeny, Tissue Localization, and Different Populations of MSCs in Tissues

MSC were first described by Friedenstein and co-workers in the 1990s as spindle-shaped cells isolated from the bone marrow (BM) in an attempt to identify multipotent stromal precursor cells. Those cells were first named colony-forming unit fibroblasts (CFU-Fs) due to their presupposed origin from the stromal compartment of the BM in connection with their capacity of adherence to tissue culture vessels and the fibroblast-like appearance of the common progeny [63]. Stromal cells derived from CFU-Fs have been identified as feeder cells for the culture of hematopoietic stem cells (HSCs) and have the ability to differentiate into several subsets of cells both in vitro and in vivo: adipocytes, chondrocytes, and osteocytes among others [63]. The potential implications of these main discoveries were initially evaluated only in the light of experimental hematology and the entire role and physiology of the MSC remained, almost today and despite an important research, unappreciated [64,65].

MSCs are principally derived from mesodermal cells (MC) [66,67]. They can also be defined as multipotent mesenchymal stromal cells encompassing a heterogenous population of cells that have the ability to proliferate in vitro as plastic-adherent cells, forming colonies, with fibroblast-like characteristics, self-renewing potential, and to differentiate in several types of mature cells from different embryologic origins: ectodermal cells (epithelial cells and neurons), endodermal cells (gut and lung epithelial cells, muscle cells), and mesodermal cells as connective stromal cells, bone, cartilage, and fat cells [68]. If MSCs were first identified and extracted from BM, they have been isolated from every type of connective tissue [69]. It is important to note that the MSCs are localized at the perivascular level in the different organs playing the role of an interface between the inner and outer environments, thus functioning as a possible gatekeeper against possible infection and yet being able to alert professional immune cells to trigger the IIR.

Although MSCs are considered of MC origin, MSCs can also derive from the neural crest (NC) ectodermal embryonic tissue [70,71]. The NC gives rise to numerous progenitors: NC-MSCs but also melanocytes, Schwann cells, thyroid C cells, and adrenergic cells of the adrenal medulla [72]. NC-MSCs have the particularity of being able to differentiate into many cell types, among which we find endoneural fibroblasts, osteoblasts, and chondrocytes [73,74]. All NC-derived progenitors (NC-Pg) have nurturing functions and aid in the education and maturation of other cell subpopulations. Some NC-Pg (e.g., melanocytes) thus have an essential nourishing function with respect to neurons and keratinocytes. NC-MSCs have educational and maturation support functions for the cell populations contained within organs as varied as the lung, stomach, skin, bone marrow, or even the immune system [71].

Perivascular MSCs (pMSCs) result from an embryologic process of migration to the wall of blood vessels around endothelial cells and next to peripheral nerves [69,75,76,77]. pMSCs are able to differentiate into collagen^high^-producing myofibroblasts contributing to kidney or liver fibrosis and express canonical glial markers as glial fibrillary acidic protein (GFAP) and myelin P zero protein (P0), surface markers of astrocytes and Schwann cells respectively. This particular population, expressing glial markers and derived from NC was consecrated as glial MSC (gMSC) [78]. pMSCs are found around the vessels of different organs. The pMSCs located within the liver, as recently emphasized by Mederacke et al., are better known as hepatic stellate cells (HepSC) [79,80].

The ontogeny of MSC deriving from either the ectoderm (neural crest, NC) or the mesoderm is shown in Figure 2.

### 2.2. MSCs Characterization and Surface Markers Expression

As aforementioned, an MSC is characterized by its origin, its multipotent stem cell capacity, and its ability to differentiate notably into lipo- or myo-fibroblasts. MSCs might also be identified by the summation of expressed selective surface markers such as CD29 (integrin β1), CD44 (hyaluronic acid receptor), CD71 (transferrin receptor), CD73 (ecto-5′-nucleotidase), CD90 (GPI-anchored THY1), CD105 (TGF-ß R or endoglin), CD140 (PDGFR), CD146 (MUC18/melanoma cell adhesion molecule), CD271 (low-affinity nerve growth factor receptor and also known as p75 neurotrophin receptor), CD248 (endosialin, tumor endothelial marker 1), nestin (type IV intermediate filament of the NC), NG2 (Chondroitin sulfate), GFAP (glial fibrillary acidic protein of astrocytes), desmin (type III intermediate filament), and ganglioside GD2 [66,78]. The relative amount of each of these surface molecules varies with the MSC tissue localization and state of activation. MSCs are also characterized by the absence of several markers including those of the hematopoietic lineages (CD14, CD34, CD45, CD19), co-stimulatory molecules (CD80, CD86, and CD40), and other immune surface markers (CD11b, CD79alpha, and HLA-DR) [81]. These surface markers have been characterized on cell populations cultured in vitro and it is commonly accepted that several markers may be lost compared to the MSCs observed in vivo.

### 2.3. Physiological Roles of MSCs: A Niche for Stem Cells and Tissue Gatekeepers Alerted in Response to Tissue Injuries

MSCs are stromal cells that can be identified in adult tissues such as BM, lung [82], heart [83], synovial membrane, periosteum, skeletal muscles, dental pulp [84], adipose tissue [85], bones, and around the vessels. It is also possible to find them in large numbers in many fetal tissues such as amniotic fluid and membrane [86], placenta [87], umbilical cord blood, and Wharton’s Jelly where they may exert important immunoregulatory functions [88,89].

The main role of MSCs has long been a subject of ongoing debates on the ground that they are ultimately and extremely pleiotropic and totipotent [90]. Their pleiotropism would suggest a potential for actions out of proportion compared to other cell types, including professional immune cells. The ability of MSCs to migrate to the site of injury, to remove the intruders and to engage stem cell recruitment makes them “master orchestrator” of tissue repair in response to diverse injuries. MSCs, notably those from the NC, act as a niche for stem cells in the BM [91,92]. They interact with other cell types present in different organs with the help of growth factors. These growth factors produced by MSC are epidermal growth factor (EGF), fibroblast growth factor (FGF), transforming growth factor (TGF-beta), vascular endothelial growth factor (VEGF), hepatocyte growth factor (HGF), insulin-like growth factor (IGF), and angiopoietin-1 [93,94].

In response to tissue injuries associated with infectious diseases, MSC expressing PRRs responds to DAMPS or “alarmins” [95,96] and PAMPs represented by constituents of pathogens (viruses, bacteria, parasites). Pro-inflammatory molecules and hypoxia which are observed during acute critical attacks such as sepsis, are vectors of upregulated expression of PRRs such as TLRs [97]. These PRRs localized at the MSC surface are capable of binding both PAMPs [98,99,100] and DAMPS (HMGB1, HSP, and DNA proteins), degradation products of host cells or pathogens. Fine regulation of the response to the presence of DAMPs/PAMPs is crucial to avoid a “runaway” phenomenon of inflammation and a risk of an inappropriate autoimmune reaction [101]. 

MSC can also regulate the recruitment of innate and adaptive immune cells. Indeed, they can produce chemoattractant molecules (e.g., C3a, ATP, lysophosphatidic acid LPA1) and chemokines (e.g., CCL2/MCP1, CCL5/RANTES) to recruit professional immune cells at the site of the injury. This will lead to the activation of innate immune cells (PNN and monocytes) which play a critical role in local inflammation by expression of diverse inflammatory factors such the migration inhibitory factor (MIF) [81]. These recruited immune cells will greatly contribute to the phagocytosis of the infectious agents as well as the host-derived cell debris, both fueling the local and systemic inflammatory responses.

The identified roles of MSCs in controlling effectors of innate and adaptive immunities represent one of the cornerstones of its implication in the pathophysiology of septic states as explained below.

An outline of the different canonical and physiological roles of the MSCs is proposed in the Figure 3.

## 3. Study of the Role of the MSCs in Sepsis/Septic Shock: What Do We Know?

### 3.1. Contribution of Knowledge of the Role of MSCs in the Comprehension of the Septic Shock: Interactions between MSCs and the Immune System

It is increasingly recognized that MSCs represent cornerstones of the immune system and possess the capacities to modulate the immune response. Although this immunomodulatory potential of MSCs is the subject of numerous studies, the interactions of MSCs with target cells and their role in sepsis are still poorly understood. This key role is provided through two of their main characteristics: (1) Their ability to express soluble proteins and surface markers of innate immunity allowing the detection, management, and destruction of pathogens; and (2) their immunomodulating capacity in inflammatory conditions avoiding overexpression and overactivation of innate immune effectors. MSCs therefore express numerous markers specific of innate immune cells which can be divided into three categories: (1) Pathogen recognition receptors (PRR); (2) pro-inflammatory cytokines and chemokines; (3) immune effectors playing a direct anti-microbial or toxic role on pathogens. 

Several authors have noticed that MSCs have the ability to regulate innate and adaptive immune responses both in vitro and in vivo [81,93,102]. MSCs are able to activate several pathways involved in maintaining the balance between proinflammatory and anti-inflammatory aspects of the immune system. More precisely, several authors noticed the capacities of MSCs to regulate the inflammatory response by activating different axes. This modulation of the IR can occur through two distinct mechanisms: (1) The secretion of soluble factors having a paracrine action such as IFN gamma, nitric oxide (NO), indoleamine 2,3-dioxygenase (IDO), prostaglandin E2 (PGE2), TGF-ß, and IL-10; (2) the direct cell–cell contact between some specific membrane receptors of MSCs and those located on the surface of target cells.

The modulation of immune responses by MSCs thus occurs through the activation of numerous mediators:-IFN-γ, alone or combined with other pro-inflammatory cytokines such as TNF, IL-1α, or IL-1β are able to induce the secretion of chemokines responsible for the activation of iNOS and the attraction of T cells. MSCs are also able to control the proliferation and activation of macrophages, natural killer T (NKT) cells, and γδ T cells [103]. MSCs, after stimulation with the inflammatory cytokines IFN-γ, TNF-α, or IL-1, can express inducible (i) NOS, releasing NO. High NO concentrations can, in turn, inhibit the transcription of (STAT)-5 phosphorylation in T cells and decrease the apoptosis of immune cells, participating in immunomodulation [104].-Indoleamine 2,3-dioxygenase, also expressed by MSCs stimulated by IFN-γ, inhibits lymphocyte proliferation by depleting tryptophan in the microenvironment. IDO-secreting MSCs are also potent inhibitors of Th1 cells and NK activity with the help of PGE2.-Role of cyclooxygenase-2 (COX2) and prostaglandins expressed by MSCs: In the context of a severe infection associated with high levels of LPS and/or host-derived factors (e.g., TNF-α) or even in hypoxic conditions, MSCs will engage the stimulation of the NFκB pathway. Activation of NF-κB signaling can upregulate the expression of COX2 and the COX2-dependent increase of PGE2 synthesis. PGE2 in turn will bind to G-protein-coupled receptors EP2 and EP4 on macrophages to increase the expression of the canonical anti-inflammatory cytokine IL-10 chiefly involved in the control of an overt inflammatory response [96,97].-Other soluble factors released by MSCs, such as IL-6, have the ability to slow down oxidative stress, apoptosis of neutrophils, and the differentiation of bone marrow progenitors into APC [105,106].-HLA-G5 is secreted by MSCs stimulated by IL-10 and following contact between MSCs and activated T cells. HLA-G5 has an anti-proliferative action on T cells, NK cells, and cytotoxic T lymphocytes.

From a cellular point of view, MSCs are known to control proliferation, IFN-γ production, and cytotoxicity of both CD4+ and CD8+ T cells [105]. Regulatory T cells (Tregs) are a subpopulation of CD4+ T lymphocytes originating from the thymus (tTregs) or from the periphery (pTregs) [107,108]. These immune cells are involved in the tolerance and homeostasis of the immune system. During septic states, these Tregs act as modulators of the immune system allowing control of the inflammatory response. Most of the experimental studies focusing on Tregs during sepsis observed an up-regulation of this population which can explain a long-term immunosuppression [109,110]. Conversely, Carvelli et al. found a decrease of these lymphocytes during septic shock [111]. If the proportion of Tregs remains to be better determined during sepsis, it appears that these cells play a key role in restoring the immune balance after the early inflammatory phase. MSCs are reported to favorably alter the Th1/Th2 balance toward Th2 cells [112] and are able to convert conventional T cells into Tregs [81]. 

It is well described that in the event of critical tissue damage in vivo, MSCs are attracted to various target sites (ischemic or injured sites) by a phenomenon of “homing” [113] mediated by the pathways of SDF1-α/CXCR4 or CD44 present on their surface and which can interact with hyaluronic acid (molecule exposed in the event of an acute lesion with attack of the connective tissue) [106,114]. MSCs possess the ability, in response to pro-inflammatory cytokines secreted upon contact with a pathogen, to produce chemokines and intercellular adhesion or vascular adhesion molecules such as (ICAM)-1 or (VCAM)-1 respectively [115]. The activation of these chemokines and cellular adhesion factors in return promotes the migration of immune cells.

Ischemia-reperfusion represents a mechanism that occurs in sepsis and septic shock conditions. It has been suggested that IL-10-overexpressing BM-MSCs could prevent lung IR-injuries in rats with a decrease of CD4(+) and CD8(+) T cells in the lung, improving blood oxygenation in the treated group [116].

Experimentally, the dynamic in vivo distribution of MSCs has been monitored in animal models. When MSCs are administered intravenously, several phases are distinguished, characterized by different MSC distributions. MSCs first migrate to the pulmonary and hepatic capillary bed [117]. They seem to reside at this level for a period extending over 96 h, before becoming undetectable within the sites considered [118]. Although this homing phenomenon to the lung and the liver remains unclear [119,120], these observations are at the origin of the theory according to which MSCs may act at a distance on their targets using mediators by two distinct pathways: (1) a paracrine pathway mediated by a direct secretion in the environment and (2) a pathway mediated by the secretion of extracellular vesicles (EVs). Whether this paracrine pathway is mediated by EVs or not, the two pathways have in common the fact that they can exert a remote action on the immune effector cells of immunity and do not require mandatory proximity of the MSCs to mediate their immunoregulatory effects.

EVs encompass several types of particles made up of membrane material formed from a parent cell. The internal content is thus protected during its transit to the target site by a lipidic bilayer which differentiates from the mother cell membrane by an enrichment in specific lipids (cholesterol, glycosphingolipids, and phosphatidylserine) allowing increased longevity within the compartments of the organism, especially the vascular environment [121,122] and direct physiological actions [123]. They can be classified according to their size and content. A distinction is thus made between apoptotic bodies derived from cells in apoptosis (1000 to 5000 nm), microvesicles (MVs) formed directly from the plasma membrane (100–1000 nm) and exosomes (Exos) formed from the endosomal system (40–150 nm). Usually, it is easy to distinguish apoptotic bodies from Exos and MVs due to the possible separation by ultracentrifugation. The EVs thus commonly designate the group of vesicles comprising the MVs and the Exos obtained after extraction. The content of the EVs, once extracted, can be analyzed. This heterogeneous EV’s material and content includes nucleic acids of which coding and non-coding RNA (mRNAs, miRNAs), lipids and proteins [124,125,126]. Among the non-coding RNAs, we can distinguish the following microRNAs (miRNAs) in MSC: miR-221, miR-23b, miR-125b, miR-451, miR-31, miR-24, miR-214, miR-122, miR-16, miR-150, and miR133b [127,128,129]. These miRNAs are involved in vitro in the mechanisms of tumor genesis, apoptosis, angiogenesis, and modulation of immunity [127,128,129]. More than 5000 different proteins derived from MSC EVs and possessing roles in self-renewal, differentiation, homing and signal transduction have so far been identified [130]. These EVs and their content may be essential players in the immunomodulatory and repair roles of MSCs. Several studies have thus been able to demonstrate the presence of cytokines such as IL-10, IL-6, IL-37, lipocalin-2, TGF-β, programmed death ligand-1 (PD-L1), and galectin-1 within these EVs [130,131,132]. The above findings are supported by studies that have observed an increase in the number of EVs released by MSCs in physiological inflammatory and stressful situations, such as sepsis [131,133,134]. Beyond their use in therapeutic trials during sepsis, the Exos derived from MSCs seem to be involved in their immunomodulatory potential during pro-inflammatory conditions and acute injuries [135]. Exos are capable of presenting antigenic motifs directly to specialized immune cells and activate CD8^+^ T lymphocytes and NK cells [136]. They are also carriers of bioactive molecules and indirect antigen presenters through modulation of the response of subpopulations of immune cells [137]. MSC-Exos finally have immunosuppressive capacities and control the activation and proliferation of a large panel of innate or adaptive immune cells such as B lymphocyte cells (LyB), NKs, LyT or LyT CD3+ and LyT CD4+ populations while preserving the activity of Tregs [138,139]. They also have, like the MSCs themselves, the capacity to down-regulate the production of pro-inflammatory cytokines such as TNF-α [140] and to up-regulate the production of anti-inflammatory cytokines (IL-6, IL-10, and TGF-β) [141]. MSC-Exos are therefore able to orient the individual’s immunity toward immune tolerance with a preferential M2 type polarization [142,143,144].

The different interactions between MSCs and immunity effector cells in acute tissue injury conditions and the hypothesis of interactions with immune and pro-inflammatory effectors in septic conditions are summarized in Figure 4.

### 3.2. MSCs Interacting with PNN, and the Key Role of the SDF-1/CXCR4/CXCR7 Pathway

Sepsis involves a very early innate immune response, in particular throughout the systemic inflammatory response to the aggressor. One of the key cells of this IIR is represented by PNN. The PNN participates in the control of the pathogen by its antimicrobial action (i.e., Release of ROS such as H_2_O_2_) in the early phase of the infection. The PNN is thus one of the first specialized cells recruited to the site of infection [145]. Its recruitment and early migration from the circulation to the site of inflammation make it a good prognostic marker predicting mortality [146]. A dysregulation of PNN functions is the cause of increased morbidity and mortality [147].

The PNN has a short lifespan and quickly moves toward cell death once its mission has been carried out using a programmed cell death mechanism called “pyroptosis” dependent on the caspase 1, 4, 5, and 11 pathways in particular [148]. PNN is continuously produced in the BM from progenitors of the granulocyte lineage and the “mature” forms are released into the systemic circulation just after the pathogen aggression [149]. During sepsis, an increase in the lifespan of these PNNs [147] as well as in the recruitment of immature forms were observed and is associated with increased mortality [150,151].

The SDF-1/chemokine (C-X-C motif) ligand 12 (CXCL12) pathway is a major pathway involved in the mechanisms of IIS. It is at the origin of the activation of multiple cellular pathways responsible for the recruitment and migration of PNNs from the production site to the site of aggression or “homing” phenomenon [152]. SDF-1 has two types of receptors: CXCR4 and CXCR7, both expressed on the surface of hematopoietic cells and in particular PNNs [153,154,155]. The SDF-1/CXCR4 pathway participates in the maintenance of the hematopoietic niche of PNNs and their release from the BM into the circulation during inflammatory and septic states [156]. Several authors have demonstrated a down-regulation of this pathway during septic states at the level of the BM contrasting with an up-regulation at the level of peripheral tissues with the consequence of chemotaxis of the PNNs toward the production sites in response to the presence of the pathogen [157,158].

Several studies have demonstrated an increase in the tissue expression of SDF-1 and its CXCR4 and CXCR7 receptors in the peritoneum, lungs, and liver in mouse models of peritonitis [159], and as confirmed in vitro [160,161]. SDF-1 and CXCR4 have also been the subject of studies showing their potential as early markers of sepsis [162]. If the involvement of the SDF-1/CXCR4/CXCR7 pathway in inflammatory and septic states is beyond doubt, its exact role in the regulation of the inflammatory balance is however unclear and is the subject of contradictory observations. Thus, the teams of Delano et al. and Guan et al. observed in experimental in vivo models that the administration of SDF-1/CXCL12 analogues increased the survival of mice in septic shock, whereas its depletion was responsible for increased mortality in individuals [158,163]. Conversely, Ramonell et al. showed that inhibition of CXCR4 reduced the mortality associated with sepsis in an experimental model of multimicrobial sepsis [164]. Gosh et al. clarified this conundrum by showing that inhibition of CXCR4 reduced cell migration by regulating the modulation of the cytoskeleton [165]. Other authors have demonstrated that inhibition of the CXCR4 pathway using pharmacological antagonists allowed protection of the lung and participated in the maintenance of tissue homeostasis during acute and chronic pulmonary inflammatory processes by reducing the CXCR4+ PNN infiltrate.

More recently the team of Kwon et al. have shown that the SDF-1/CXCL12 pathway activation tightly involves MSCs and results not only in the recruitment of PNNs from the BM, but also in increased phagocytotic activities of mature and immature PNNs [166]. The PNN plays a central role in circumspection of infection. If the infection persists, mature PNNs will no longer be sufficient and the SDF-1/CXCL12 pathway activation then participates in the recruitment of immature PNNs [158]. Although the bactericidal and phagocytotic capacities of these PNNs are less pronounced [167], they participate in the fight against the invading pathogen. These phagocytic capacities are significantly diminished when this pathway is silenced [166].

Finally, the activation of this pathway is at the origin of the synthesis of pro-inflammatory cytokines and is involved in the integrity of the endothelial cell barrier through the MAP-kinase (MAPKs) pathway and the transcription of NFκB p65 [168]. More recently, Ngamsri et al. have demonstrated the involvement of the SDF-1/CXCR4/CXCR7 pathway in cell-tissue interactions via tight junctions in experimental models of peritonitis and associated sepsis. Thus, blocking this pathway makes it possible to restore the integrity of the endothelial barrier, to reduce tissue edema and its consequences by a mechanism dependent on the A2B adenosine receptor [159].

### 3.3. Circulating MSCs and Sepsis: Toward a Novel Entity?

It is currently well established, and this for about ten years, that MSCs can be observed in vivo around vessels (arterioles and arterial and venous capillaries). These MSCs, conveniently called pericytes, maintain contact with the basement membrane and are separated from the vessel lumen by endothelial cells.

The ability of MSCs and in particular of pMSCs to evade and gain access to the blood circulation (see Figure 4, ③) in vivo remains ill-characterized. Several works denoted the impossibility of detecting MSC in the circulation [169]. In the same way, authors have tried to obtain, without success, cultures of MSCs from blood puncture from the portal vein, a technique allowing to avoid peripheral tissue contamination [66]. Kuznetsov et al. have successfully isolated, in minute quantities, circulating MSCs from four different mammalian species, capable of donating osteogenic cells and adipocytes [170]. Other studies were able to individualize MSCs in the circulating blood under stimulation conditions by G-CSF or VEGF [171,172,173]. Likewise, searching for circulating MSCs in injured mice have allowed the isolation of larger quantities of these cells, associated with increased circulating levels of G-CSF and VEGF. Given the literature, two facts are interesting to note: first, obtaining circulating MSCs seems easier under stimulatory conditions and, second, MSCs from the blood of individuals subjected to these conditions, appear to have a greater differentiation potential in culture than those obtained from peripheral blood from healthy individuals. The capacity for contraction and mobilization of pMSCs deeply in the vascular bed seems to be related to several cellular pathways independent of the iNOS pathway in mouse models of sepsis [174]. Of critical note, circulating MSCs have recently been found in patients suffering from rheumatoid arthritis (RA) or from cancer [175,176]. Blood transcriptional profiling experiments indicated that, in RA, B-cell autoimmune activation was followed by expansion of circulating podoplanin + MSCs (but negatives for myeloid CD45 and endothelial CD31 markers) corresponding to pre-inflammatory mesenchymal, or PRIME, cells in the blood of patients [175]. This finding has exciting and important implications from a biomarker point of view and also from a pathological standpoint given that these circulating MSCs may recapitulate secondary inflammatory-autoimmune sites, distant from the primary sites, a phenomenon which is well-known in cancer metastasis particularly as a side effect of irradiation [176]. Of note, a recent study using FACS analysis revealed increased numbers of circulating MSC (CD29+ CD73+) in the blood of patients with ARDS [177]. High levels of PCT and basal blood levels of VEGF and angiopoietin two were observed in these patients, arguing that the latter two molecules should not be considered as the mobilizing factors.

These different works lead us to formulate the hypothesis that pMSCs must play a dual role after recirculation: (1) an immunomodulatory role in the blood stage after leaving their perivascular niche in response to the systemic injuries. The migrating phenotypes may be associated to high levels of PAMPs. (2) A trophic role while responding to host cell debris (DAMPs) to help to repopulate the damaged tissue with stem cells. The immunomodulation phase in blood would concern the pMSCs co-responding to specific pro-inflammatory cytokines and chemokines produced by circulating and activated immune cells. This migration should involve specific set of adhesion-ligand molecules uniquely expressed at the basal membranes of the endothelium. When returning to tissue bed and in essence recapitulating their embryonic behavior (NC-like), they are likely to affect the perivascular niche with two possible scenarios: 1. They will recruit stem cells (including those from the BM) but also, 2. favor immune cell recruitment. In severe inflammatory conditions, the scenario 2 may contribute to autoimmune flare as recently described for the PRIME cells in RA patients. The unanticipated role of B cells and derived factors in this process needs to be explored further.

### 3.4. Diagnostic and Therapeutic Consequences

If these different hypothesis prove to be correct, MSCs could play a major role in the genesis of septic shock. Thus, septic shock could be the consequence of several isolated or intricate phenomena: (1) a dysfunction of the pMSCs allegedly involved in immunoregulatory activities but which may reveal a more aggressive phenotype while being harassed by the myriad of inflammatory factors. In light of their past education program while being derived from the NC, MSCs have a unique relationship with endothelial cells and peripheral nerves. For unclear reasons, when they are dissociated from this cell–cell communication settings, they are uncontrolled and behave abnormally. This original paradigm makes us argue that returning to homeostasis should aim to restore the crosstalks between pMSCs and endothelial cells as well as pMSCs and nerves. pMSCs express several receptors for neurotransmitters and provide opening new pharmacological therapeutic avenues. From a therapeutic point of view, the identification of different cellular profiles of pericytes, pMSCs, and of circulating activated MSCs would improve our physiological knowledge and could pave the way for new molecules targeting these cellular pathways in the treatment of septic shock.

From a diagnostic standpoint, the critical involvement of MSCs in the processes leading from sepsis to septic shock could result in very early changes in MSCs circulating in the blood. It would be interesting to perform kinetics of the circulating levels of these MSCs during sepsis in order to identify the unique disease state patterns. The use of RNA sequencing as performed for the PRIME studies will help to associate the presence of these circulating MSCs with a unique “mesenchymal” RNA signature in blood samples. This type of analysis will benefit from the isolation of exosomes-derived from MSCs prior to the molecular analyses.

## 4. Perspectives for the Study of MSCs in Sepsis Pathophysiology

### 4.1. Use of MSCs as In Vivo Immunomodulators

The natural history of sepsis and the possible rapid onset of septic shock make it difficult to resort to an administration of autologous MSCs conditioned in vitro, because of a limited time for clonal expansion. The possibilities offered by the transfusion of allogeneic MSCs, from healthy subjects, are thus the basis of more therapeutic trials. Allogeneic MSCs also have the advantage of being poorly immunogenic in vitro [178], facilitating their tolerance by the host, both in preclinical [179] and clinical studies [96]. Nevertheless, the less immunogenic nature of allogeneic MSCs [180,181] raises the question of their use in the context of sepsis, in particular, due to major inter-individual phenotypic variations which may interfere with their immunomodulatory capacities depending on the inter-cellular contacts that they may encounter while being in the blood.

#### 4.1.1. Preclinical Data and Control of Inflammatory Processes in Experimental Studies

For the past ten years, many authors have shown interest in the therapeutic contribution of MSCs to treat critical illness or organ failures in the context of inflammatory or septic aggression. Numerous preclinical data were able to show that MSCs can have a protective action in murine models of sepsis, modify the potential evolution of sepsis toward a septic shock [182], and reduce its consequences through their immunomodulatory, anti-bacterial, and anti-inflammatory properties as well as their restorative potential after elimination of the initial threat. The time window for the injection of the MSCs is important and aims, according to our hypothesis, to have low levels of circulating PAMPs before the injection. It remains to be tested whether antibiotics may directly affect MSCs immunoregulatory activities.

The systemic administration of allogeneic MSCs in a mouse model of acute renal failure has made it possible to improve the recovery of renal function in association with an inhibitory effect on the production of pro-inflammatory cytokines (Il-1β, TNF-α, and IFN-γ) and an anti-apoptotic effect on kidney cells [183]. A study involving the injection of MSCs in a model of pulmonary fibrosis showed an effect on the secretion of the IL-1 receptor antagonist (IL-1RA), inhibiting Il-1α-producing T cells and TNF-α producing macrophages [184]. Parekkadan et al. have also observed the anti-inflammatory and immunomodulatory roles of conditioned MSCs in an experimental model of fulminant hepatic failure, in connection with a reduction in leukocyte proliferation and infiltrate and apoptosis of hepatic cells [185]. MSCs also have a protective effect against cerebral deleterious consequences of sepsis by controlling neuroinflammation. Several studies have thus been able to show the protective role of MSCs in mouse models of autoimmune encephalitis by inducing immune tolerance by the targeted inhibition of myelin-specific T cells [102] and by inhibiting the expression of anti-myelin autoantibodies by T cells, preserving axonal capital [186]. While many authors have found an antagonist effect of the administration of MSCs on the secreted pro-inflammatory factors [187,188], other studies had different results, showing no impact on the inflammasome, especially in porcine models [189]. Despite sometimes discordant results, meta-analysis are available on the subject, highlighting an overall downward trend in the mortality of individuals with septic conditions, after administration of MSCs [190].

These data and a selection of studies providing preclinical data on the administration of MSCs during sepsis are summarized in Table 1.

#### 4.1.2. Infusion of Derived-MSCs and Clinical Studies of Sepsis Models

Many works have focused on testing the infusion of MSCs during inflammatory conditions complicating septic states, with acute respiratory distress syndrome (ARDS) in the foreground. In contrast, we only find a few clinical trials including MSCs in septic states strictly speaking in the literature. Most are phase I to II trials focusing on the safety of injecting MSCs in sepsis and septic shock using allogeneic MSCs derived from adipose tissue, BM or UC harvest (UC-MSC). A first phase I study dating from 2017 looked at the safety of injection of MSCs in septic shock. A “phase I/II” clinical trial (NCT01849237) has been conducted in neutropenic patients (<1000 PNN/mm^3^) with an administration of doses of 1 to 2 million MSC/kg/day started within 10 h of the onset of the septic shock. This trial, which is currently underway, is looking at D28 mortality, the reversibility of septic shock, the extent of organ dysfunction and the impact of MSCs on the biological parameters of inflammation [191]. Another phase I study has compared MSC infusion in increasing doses (1 × 10^6^, 2 × 10^6^ and 3 × 10^6^ cells/kg) in 15 patients with severe sepsis [192]. Finally, a phase I trial (NCT02328612) analyzes the impact of infusion of MSCs derived from adipose tissue on the systemic inflammatory response in healthy patients aged 18 to 35 years in a model of inflammation by injection of lipopolysaccharide (LPS) and attempts to determine the optimal dose of MSCs to be administered for the control of inflammation (four groups: placebo, groups of 0.25 × 10^6^ cells/kg, 1 × 10^6^ cells/kg, and 4 × 10^6^ cells/kg) [193].

The preclinical data also led to the realization of clinical trials which confirm the decrease of pro-inflammatory states complicating septic states and the associated organ failure. The START trial (STem cells for ARDS Treatment, NCT01775774) was a multi-center, randomized, controlled trial that tested a single dose of intravenous BM-MSCs-type MSCs in the treatment of moderate to severe ARDS [194]. The lung injury score (LIS) and sepsis-related organ failure assessment score (SOFA score) were lower at H72 in the group receiving high doses of MSCs although these differences were not significant (*p* = 0.87 and *p* = 0.76 respectively). The authors observed very good tolerance in the different groups without any serious adverse events related to the treatment being able to be highlighted.

More recently, two studies have focused on therapeutic infusion of MSCs during ARDS complicating SARS-CoV-2 infection. The first study was conducted on a 65 years unique woman, treated by allogenic human UC-MSCs at three different time-points (5 × 10^7^ cells each time) with a remission of the inflammation symptoms showed by laboratory indexes and CT images attributed to the injection but occurring on the 16th day after the diagnosis, which may correspond to the natural course of the disease [195]. A second study tested the injection of MSCs in 7 patients with an improvement of inflammation parameters but with very heterogeneous severity groups, limited follow-up, and without a control group [196].

Most of the clinical studies conducted using MSCs have many limitations: they relate to small numbers, show large disparities in the design adopted and in the associated biomarker analyses, and probably lack standardization in the selection of types of sepsis. In addition, many clinical studies concern states of systemic inflammation such as ARDS or ALI and few have been carried out in septic patients per se. Finally, the vast majority of these studies are phase I clinical trials. The various clinical trials concerning the use of MSCs in the context of inflammatory and septic states are summarized in Table 2.

While most of these trials confirm the safety of administration of allogeneic MSCs, there are very few results available from phase II trials. A meta-analysis by Can et al. published in 2017 grouping together more than 90 phase I clinical trials investigating the use of MSCs derived from umbilical cord harvesting also reported excellent tolerance of these IV infusion with minimal adverse effects marked by pain at the injection site and rapidly resolving flu-like symptoms [201]. The various clinical studies published to date on the use of allogeneic MSCs for the treatment of inflammatory and septic conditions are summarized in Table 2 and classified according to their type (phase I or II trial) and the pathophysiological characteristics of the populations studied.

Among the studies of interest concerning sepsis not yet published, we can cite the phase I CISS trial carried out in open label using allogeneic BM-MSCs (NCT02421484) in three dose cohorts with three participants per cohort who received allogeneic BM-MSCs at doses of 0.3 × 10^6^ cells/kg, 1 × 10^6^ cells/kg, and 3 × 10^6^ cells/kg from the lower dose to the higher dose. Another phase Ib/IIa, randomized, double-blind, placebo-controlled study (SEPCELL N°2015-002994-39) looked at the injection safety of AD-MSCs (injections on D1 then D3 of admission) for the treatment of several bacterial pneumonia acquired in ICU-patients [204].

#### 4.1.3. Cell-Free Based Therapies: MSCs Extracellular Vesicles and Paracrine Factors Incriminated

As aforementioned, EVs group together the endosomes and MVs of MSCs which contain proteins, mitochondrial material (a major DAMP), and a myriad of miRNAs. Numerous pre-clinical studies have demonstrated the beneficial role of the administration of MSCs exosomes in tissue repair processes (cartilage, skeletal muscles), angiogenesis, modulation of the immune response and acute or chronic inflammatory states [205,206,207,208,209]. In addition to these pleitropic effects linked to the “cargo” function of the EVs of MSCs, allowing the transport of mediators with a direct or indirect role (inter-cellular communication) on the target tissues, EVs own a preponderant immunomodulatory action, at the image of the cells that secrete them. Exosomes thus modulate the activity of many effectors of the immune response such as LyT, macrophages (M1 and M2), NK cells, and Treg as well as microglia and dendritic cells [210,211,212,213,214]. These pre-clinical studies gradually led to the use of EVs in clinical studies. 

The administration of EVs has also been tested as a therapy. Many clinical trials dealing with tissue repair, the control of autoimmune diseases, cancers, as well as cardiovascular diseases are available in the literature and their main results have been well summarized in the review by Lee et al. published in 2021 (revue Lee J Clin Med). However, few clinical trials have looked at the potential benefit of the administration of MSC-derived EVs during systemic inflammatory states and septic states. A clinical trial (NCT04356300), the results of which have not yet been communicated, used exosomes derived from UC-MSCs in the treatment of MODS after cardiovascular surgery. MVs obtained from the ultracentrifugation of cultures of human BM-MSCs were thus compared to the injection of MSCs in the control of pulmonary sepsis due to *Escherichia coli* in mice. Unfortunately, no statistical difference concerning survival and bacterial clearance was noticed between the two groups [215]. A prior infusion of the culture media of MSCs with an anti-CD44 antibody impaired the survival of mice. This result is not surprising given that the EVs obtained from BM-MSCs express canonical markers (CD90, CD44 and CD73) and are negative for CD34 and CD45 (hematopoietic markers) [216]. Thus, the use of anti-CD44 could interfere with the extracellular transport of MVs. Another mean of overcoming the drawbacks associated with the administration of MSCs (potentially carrying viral materials and/or progenies) is to administer soluble factors such as TSG-6, FGF-7, KGF, PGE-2 and other components such as non-coding RNAs (lncRNAs, circRNAs, and miRNAs). The available pre-clinical and clinical data regarding the use of MSC EVs has, however, recently been expanded considering the COVID-19 pandemic. Many phase 1 or 2 clinical trials using MSCs-derived EVs from sources as varied as bone marrow, adipose tissue, umbilical cord, are being recruited. Completed clinical studies, having used EVs administered by intravenous or inhaled route, seem to show a safety of their administration. A study published in 2020 already suggested a significant association between the administration of Exos and the improvement of the PaO_2_/FiO2 ratio (*p* < 0.001), a decrease in the absolute number of neutrophils [mean reduction 32% (*p* < 0.001)], an increase in CD3+, CD4+, and CD8+ LyT by 46% (*p* < 0.05), 45% (*p* < 0.05), and 46% (*p* < 0.001), respectively. They also found a reduction in CRP, ferritin, and D-dimer levels of 77% (*p* < 0.001), 43% (*p* < 0.001), and 42% (*p* < 0.05) [217]. The main trials using MSCs-derived EVs in COVID-19 have been described in a review recently published by Krishnan et al. [218].

### 4.2. Biomarkers of Disease States in Sepsis Including Those Related to Immune Cells and MSC Behaviors

The ability to diagnose sepsis but above all to predict the progression to septic shock remains a medical grail. To this end, many authors have attempted to demonstrate early biomarkers of these states of acute aggression. In a review, Pierrakos et al. studied 34 biological markers of sepsis including 16 for its early diagnosis. No molecule was found to have sufficient sensitivity (Se) and specificity (Sp) for routine use [219]. Thus, the study of pro-inflammatory cytokines (TNF-α, IL-6, and IL-1) has proved to be deceiving in this indication with low Se and Sp [220]. The CRP, which reflects the acute phase of inflammation, increases too late compared to the onset of sepsis (4–6 h delay) and has very poor Sp [221,222].

Other biological markers of the “ligand-receptor pair” type intervening early in the processes of adaptive immunity have also been suggested as good candidates. TREM-1 and TREM-2 (triggering receptors expressed on myeloid cells) are transmembrane glycoproteins of the immunoglobulin superfamily (Ig-SF) and lectins acting as receptors “inhibitors” of the IIR. They are constantly engaged by the presence of ubiquitous endogenous molecules (sialic acid, MHC-I, CD200) and activate tyrosine phosphatases which restrain stimuli in favor of this immune response. TREM-1 is found on the surface of PNNs, macrophages, and monocytes. The activation of TLRs during sepsis induces an overexpression of pro-inflammatory cytokines (TNF-α and IL-1α), of the membrane form of TREM-1 as well as the secretion of a soluble form of TREM-1 (sTREM- 1). This role of amplifier of the inflammatory response has moreover been confirmed in vivo in a murine model of septic shock, where the blocking of TREM-1 reduced the overall mortality. The determination of the sTREM-1 seems to be more interesting with an AUC of 0.91 (95% CI 0.88–0.93) in a meta-analysis involving 980 patients [223]. The measurement of the expression of CD64 (Fc gamma receptor), a surface marker of PNN highly induced by sepsis is also found to be efficient with an AUC of 0.94 (95% CI 0.92–0.97; *p* < 0.001) for the discrimination of septic patients [224]. The early determination of CD64 combined with the SOFA score also appears to have good area under curve (AUC) for the early diagnosis of sepsis in emergency medicine [225]. However, these results concerning the last two markers require confirmation by interventional studies. 

The presepsin or soluble CD14 subtype (SCD14-ST) is a soluble form isotype of the GPI-anchored molecule of macrophages and monocytes serving as a co-receptor for the LPS-lipopolysaccharide binding protein (LPS-LPB) and TLR4; it is currently recognized as a marker for sepsis [226,227]. suPAR is a soluble form of uPAR which is a urokinase-type membrane receptor plasminogen activator receptor. Its concentration is correlated with the activity of the immune system and is also stable under physiological conditions [228,229]. Presepsin and suPAR are also of interest for the progressive prognosis of sepsis but also have many limitations [230,231]. 

Other works have studied the dosage of endothelial progenitors cells (EPCs), derived from the BM, as early markers of sepsis and its severity, finding circulating levels of EPCs (cEPCs) statistically higher in septic patients than in healthy subjects (45 +/− 4.5% vs. 12 +/− 3.6%, *p* < 0.001) [232]. These levels of cEPCs were higher in surviving patients than in nonsurvivors, defined as death within 28 days after onset of sepsis (*p* < 0.0001) [233].

Procalcitonin or PCT, a precursor peptide of calcitonin, is secreted ubiquitously in response to microbial infection consecutively to an up-regulation of the CALC-I gene in response to stimulation by pro-inflammatory cytokines. Thyroid cells (derived from the NC embryonic tissue) have high expression levels of PCT. Its kinetics are more interesting than CRP because its serum concentrations, <0.1 μg/mL in healthy subjects, increase as early as 4 h after the onset of sepsis with an earlier peak [222,234]. Its usefulness in the discrimination of sepsis, proven, is undoubtedly more linked to its good negative predictive value (NPV) and its sequential dosage in intensive care units for the diagnosis of secondary infections. It is also increasingly used as a marker for the follow-up of the effectiveness of antibiotic therapy and its early de-escalation [222]. In a meta-analysis of 30 studies and 3244 patients, PCT had a mean sensitivity of 0.77 (95% CI 0.72–0.81) and specificity of 0.79 (95% CI 0.74–0.84) and the area under the receiver operating curve (ROC) was 0.85 (95% CI 0.81–0.88) [235]. Cultured adipocytes have been shown to secrete PCT in response to the addition of IL-1β [236]. The levels of PCT secreted seem to depend on the type of bacteria found to cause sepsis (more important in Gram-negative infections, especially in obligate anaerobes) [237]. This observed difference in concentrations is explained by the different levels of stimulation linked to the polymorphism of PAMPs and DAMPs, responsible for different cytokine profiles depending on the type of aggression [238]. The addition of IFN-gamma in adipocytes culture, a molecule secreted in particular by helper T lymphocytes during viral infections, has the effect of blocking the synthesis of PCT [236]. IFN-gamma therefore has a negative biofeedback effect on the induction of PCT synthesis, which also explains the lower levels of PCT observed during invasive fungal infections which are also accompanied by IL-17 secretion with an effect similar to IFN-gamma [239]. While the mechanisms that lead to the secretion of PCT seem well understood, a crucial question remains: which cell types are responsible for this secretion? While it is well established that this is ubiquitous in vivo and that cell populations of cultured adipocytes can synthesize it, it is important to note that no work provides a clear answer on the subject.

However, some data allow us to define avenues concerning the sources of PCT in vivo. It has thus been established that human adipose tissue is a major player in inflammation and is involved in the elevation of circulating levels of PCT levels during sepsis [240]. The lipolytic effect of the calcitonin-gene (CT-gene) related peptide and adrenomedullin are moreover involved in the metabolic processes encountered during these physiological conditions. In a 2005 study investigating the expression of calcitonin-derived peptides by quantification of their respective transcripts in RT-PCR, the authors noted a significant production of PCT from AT-MSC obtained by adipose tissue biopsies of patients subjected to IL-1β and stimulation with LPS but not from undifferentiated MSCs from the same donors [236]. PCT may therefore well be secreted by subpopulations of specialized MSCs (particularly the lipofibroblast-like MSC subset, see Figure 2), and in other tissues where MSCs are present. It is interesting to note that some authors have been able to identify the presence of PCT within specific extracellular vesicles enriched with GM1 ganglioside isolated from the plasma of pregnant patients presenting pre-eclampsia, a pro-inflammatory placental disease characterized by a dysimmunity [241]. However, this type of EVs is isolated by its affinity to the B chain of cholera toxin (CTB). These CTB-EVs are the only true exosomes derived from endosomes secreted by MSCs and which may suggest a major role of these activated MSCs in the increased levels of circulating PCT during sepsis [242].

### 4.3. Interest of the Study of miRNA in Septic States

The septic syndrome is associated with a profound alteration of the genome and its expression within the cells of the various tissues of the host [243]. Among the possible actors involved in these changes in the genome, we can distinguish the class of non-coding RNAs. This class of ribonucleotides is recognized by a growing number of studies as being involved in many biological processes including innate immunity, apoptosis, and the regulation of mitochondrial function [244,245,246]. miRNAs are a class of RNA molecules of 21 to 25 nucleotides, acting as post-transcriptional regulators of gene in cells of healthy or damaged tissues [247]. miRNAs are known to play a major role in the regulation of pathologies linked to the immune system due to their capacity to modulate the expression of genes responsible for the secretion of pro-inflammatory cytokines such as TNF-α and interleukin IL-1β but also proteins linked to major intracellular communication pathways (MAP kinase pathway) [245,248,249,250]. The studies currently available concerning the detection and the assay of the non-coding RNA during the septic states relate primarily to their role in the immune processes and their potential as biomarkers. A meta-analysis studied the involvement of miRNAs in sepsis and found great heterogeneity in the themes studied and their interest as diagnostic or prognostic markers [251]. For the time being, the use of regulatory RNAs as biomarkers, although mentioned for more than ten years, concerns only a few works and limited set of miRNAs. These studies found a wide variety of miRNAs that can differentiate between inflammatory and septic states, but none has been validated. The prediction of morbidity and mortality and the risk of complications (organ dysfunctions) associated with septic patients by the detection of specific miRNAs is still in its early stages, even if miR-574-5p and miR-155 seem interesting for this purpose.

## 5. Conclusions

Despite many advances observed in intensive medicine since the 20th century, septic pathology and septic shock remain burdened with significant morbidity and mortality. The recent decision-making algorithms and recommendations issued by the sepsis-surviving campaign are struggling to obtain clear results in terms of survival for sepsis patient. The study of MSCs, which began about fifty years ago in the field of cell therapy, opens up new avenues in the understanding and control of this clinical entity. Although their therapeutic use in intensive care medicine, related to their immunomodulatory capacities, is at the center of ongoing interests, hence studies on their diverse interactions with the different effectors of immunity in vitro and during septic states must be continued. The use of cellular material derived from these cells such as MSC-derived extracellular vesicles for therapeutic purposes, or the study of miRNAs in vivo are thus good examples of therapeutic alternatives to the infusion of MSCs during sepsis. Likewise, the exploitation of their role in sepsis could lead to discoveries opening the way to diagnostic and prognostic models of septic shock. While much progress must be made before the routine use of these stem cells can be considered, further investigations on their overall physio pathological roles in diverse acute and chronic disease settings will be of great value.

## Figures and Tables

**Figure 1 ijms-23-09274-f001:**
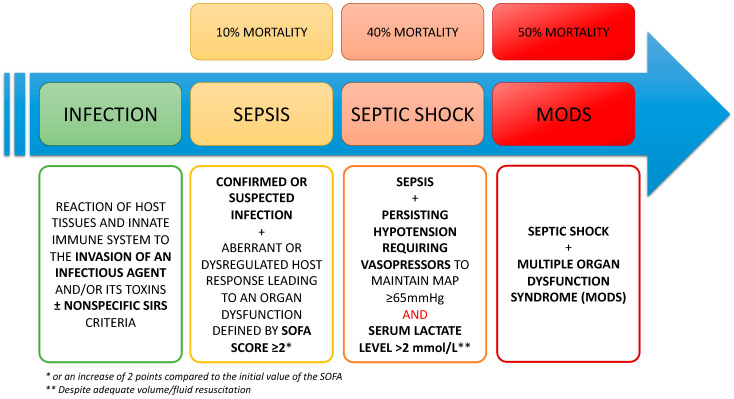
Diagram representing the continuum between infection by a pathogen and the occurrence of sepsis complicated or not with a state of shock or even a picture of multi-organ failure and their respective definitions in accordance with the recommendations of the task force and from the survival sepsis campaign. MAP: mean arterial pressure; SOFA score: Sepsis-related Organ Failure Assessment Score.

**Figure 2 ijms-23-09274-f002:**
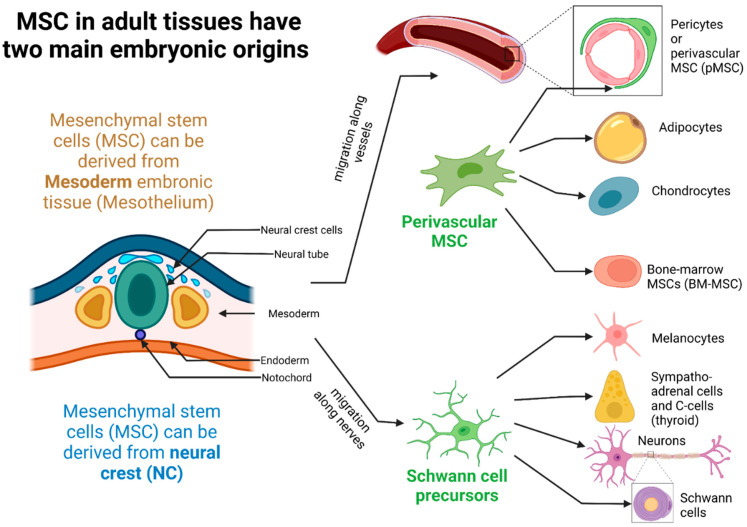
MSC originate essentially from the mesoderm or the ectoderm (neural crest-NC) embryonic tissues. Differentiated MSC notably of the NC are known to contribute to the peripheral nervous system (and the myelin-forming Schwann cells). They also contribute to important regulatory activities in response to environmental stress and for example to protect the skin from the toxic UV irradiation (role of melanocytes producing melanin pigment to protect keratinocytes). A pool of MSC from either the mesoderm or NC will migrate along blood vessels and will remain associated to endothelial cells later in life. These perivascular MSC form for instance the so-called bone-marrow stem cell niche but they are also the main gatekeepers in all major organs in adults. MSC (at least in culture) are known to differentiate into adipocytes, osteoblasts, or chondrocytes. This differentiation potential has been linked to different pathological settings whereby MSC may be involved either in tissue fibrosis (MSC differentiating into collagen-high producing myofibroblasts), in vessel calcification (osteoblast-like cells) or in fat-high producer adipocyte-like cells involved in atherosclerosis. MSC contribute to the tumor microenvironment while differentiating into cancer-associated fibroblasts (CAF).

**Figure 3 ijms-23-09274-f003:**
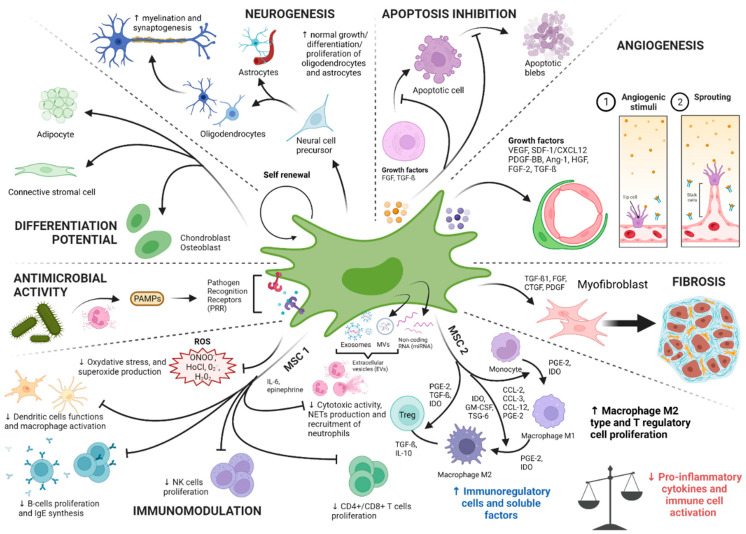
Representation of the different physiological roles attributed to MSCs according to their locations within different organs and their interactions with specific tissue-resident cell subpopulations. There are six major physiological roles associated with the MSC: (1) The capacity for neurogenesis represented by the potential for regeneration of myelin and synapses (pruning) and for the genesis of different neuronal and glial cell types; (2) control of apoptosis mediated by soluble mediators; (3) angiogenesis mediated by the secretion of numerous growth factors (e.g., VEGF, Angiopoietin) allowing the construction of nee-vessels and the repair of vessels damaged during tissue attacks; (4) anti-microbial activity by secretion of specialized proteins exerting a direct toxicity on the pathogens such as hepcidin, ß-defensin-2, and LL-37 (cathelicidin hCAP18); (5) the capacity immunomodulation and regulation of the various cellular actors of the immunity by modulating their activation, their proliferation/growth or their differentiation either by direct contact cell- cell either using soluble factors (cytokines, chemokines and non-coding RNAs) exported into the extracellular medium using EVs; and (6) the self-renewal potential of MSCs and their multipotent stem cell character which can lead to the formation of several cell types depending on the conditions of the medium in vivo and in vitro. Ac, astrocyte cells; Exos, exosomes; M1 and M2, macrophages type 1 and 2; MVs, microvesicles; Nc, neuronal cells; NK, natural killer cells; Oc, oligodendrocytes; PRR, pathogen recognition receptor; ROS, reactive oxygen species; TGFß, transforming growth factor ß; TLR, Toll-like receptor; Treg, regulatory T cell.

**Figure 4 ijms-23-09274-f004:**
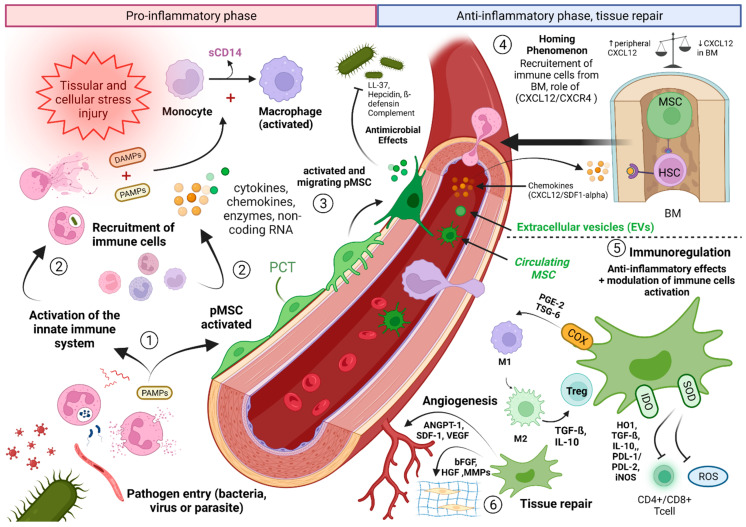
Illustration of the crosstalks between MSCs and cells of the innate and acquired immune systems during the development and the resolution of septic shock in the host. ① The inflammatory response and activation of the innate immune system at the site of the initial injury allow differentiation and activation of resident immune cells (e.g., macrophages) as well as perivascular MSCs (pMSC), both expressing a myriad of pattern recognition receptors for pathogens (PAMPs) highly conserved motifs (e.g., LPS or nucleic acids for viruses). ② Both cell types will be activated and release factors such as chemokines, cytokines, and growth factors to attract and activate blood-derived innate and adaptative immune cells. Interestingly, pMSC are known to produce the pro-calcitonin (PCT) hormone which is an early marker of the infection (bacterial >> virus) long before the liver acute phase response exemplified by the rise in C reactive protein (CRP) levels. ③ Another biomarker of sepsis, the so-called presepin molecule is the soluble form of the GPI-anchored CD14, coreceptor for LPS and known to be associated with TLR4. The appearance of sCD14 may result from the acute differentiation of circulating monocytes CD14^high^/CD16^low^ into CD14^low^/CD16^high^ (hence releasing CD14) tissue infiltrating cells. Immune cells such as neutrophils and activated MSC can release several bactericidal proteins such as LL37 as well as proteins of the complement system. The latter will contribute to pathogen opsonization, a process in Greek which means “to make the target more appetizing” and that also leads to the formation of the lytic membrane attack complex (MAC, C5b9). MSC but not pathogens will be protected from complement attack on the ground that they express high levels of GPI-anchored regulators (CD55/DAF and CD59/Protectin). ④ pMSC notably derived from the neural crest (associated to vessels and nerves in the bone marrow (BM)) play a critical role in maintaining the hematopoietic stem cells (HSC) in an immuno-privileged niche. For this purpose, MSC of the BM express high levels of the stromal-derived factor 1 (SDF1a/CXCL12) retaining HSC expressing the chemokine receptor CXCR4 (see text). Higher concentrations of CXCL12 produced by pMSC at the site of injury in sepsis will lead to a chemokine gradient in favor of HSC migration in inflamed peripheral. ⑤ pMSC as well as a little-known blood circulating MSC pool (activated in response to PAMPs, DAMPs and immune cytokines (e.g., IFN-gamma produced by T and NK cells)) will be endowed with important immunoregulatory functions (cell-cell contact mechanisms or through the release of exosomes containing regulatory miR and anti-inflammatory cytokines). ⑥ With the ultimate aim to repair the injured tissue, pMSC are well known to release growth factors to drive angiogenesis (VEGF) and/or fibrosis (TGF-β1). Fibrosis is a natural response of tissue healing and associated with the production of extracellular cellular matrix (ECM) proteins (together with matrix metallo-proteases, MMPs) such as collagens. Immune cells and notably polarized M2 anti-inflammatory macrophages are also capable of releasing these growth factors to further contribute to the return of tissue homeostasis.

**Table 1 ijms-23-09274-t001:** A non-exhaustive list of cell therapy MSC-based studies exploring the effectiveness of genetically modified-MSCs infusion in experimental animal models of sepsis or septic shock.

Indications	MSC Type	Animal Model	Results	References
ARDS	BM-MSCs	Murine model	MSCs-mediated inhibition of TNF-alpha, IL-1alpha, and IL1RN mRNA in lung, IL1RN protein in bronchoalveolar lavage (BAL) fluid, and trafficking of lymphocytes and neutrophils into the lung.	Ortiz et al. [184]
Ischemia/reperfusion Acute Renal Failure	BM-MSCs	Murine model	Beneficial effects of MSCs characterized by a reduction of the expression of proinflammatory cytokines and an up-regulation of anti-inflammatory cytokines primarily mediated via complex paracrine actions and not by their differentiation into target cells.	Tögel et al. [183]
Fulminant Hepatic Failure (FHF)	BM-MSCs	Murine model	MSCs can provide a significant survival benefit in rats undergoing FHF. The authors observed a cell mass-dependent reduction in mortality that was abolished at high cell numbers indicating a therapeutic window. Histopathological analysis of liver tissue after MSC treatment showed dramatic reduction of panlobular leukocytic infiltrates, hepatocellular death, and bile duct duplication.	Parekkadan et al. [185]
Experimental Autoimmune Encephalomyelitis (EAE)	BM-MSCs	Murine model	Immunoregulatory properties of MSCs interfere with the autoimmune attack during EAE inducing an in vivo state of T-cell unresponsiveness occurring within secondary lymphoid organs.	Zappia et al. [102]
Autoimmune Encephalomyelitis	UC-MSCs	Murine model	MSC-treated mice showed a significantly milder disease and fewer relapses compared to control mice related to a lower number of inflammatory infiltrates, a reduced demyelination and axonal loss. In vivo, PLP-specific T-cell response and antibody titers were significantly lower in MSC-treated mice.	Gerdoni et al. [186]
Sepsis and septic shock	AT-MSCs	Murine model	MSCs-immunomodulatory capacities decrease tissue inflammation by regulating cytokine homeostasis and decreasing the traffic of immune cells into organs. They own antibacterial capacities mediated by direct action on the bacterial load through secreting antibacterial peptides and by indirect action through increasing the phagocytic activity of macrophages and neutrophils. MSC infusion reduced organ failure and mortality associated with sepsis and septic shock.	Laroye et al. [182]
Septic shock	BM-MSCs	Porcine model	UC-MSCs infusion reduced peritonitis-associated hypotension, hyperlactatemia, and multiple organ failure. Cardiovascular failure was attenuated, as attested by a better mean arterial pressure and reduced lactatemia, despite lower norepinephrine requirements. UC-MSCs improved survival (60% survival vs. 0% at 24 h).	Laroye et al. [187]
Sepsis	BM-MSCs and WJ-MSCs	Porcine model	MSCs regulated leukocytes trafficking and reduced organ dysfunction. WJ-MSCs improved bacterial clearance and survival.	Laroye et al. [188]
Sepsis	BM-MSCs	Porcine model	BM-MSCs IV administration was well-tolerated. MSCs were not capable of reversing sepsis-induced disturbances in multiple biological, organ, and cellular systems.	Horak et al. [189]
Sepsis	Various types of MSCs	Various animal models	There was a statistically significant association between MSC therapy and lower mortality in sepsis animal models, supporting the potential therapeutic effect of MSC treatment in future clinical trials.	Sun et al. [190]

Abbreviations. MSCs: mesenchymal stem cells; ARDS: acute respiratory distress syndrome; AT-MSCs: adipose tissue-derived MSCs; BM-MSCs: bone marrow-derived MSCs; UC-MSCs: umbilical cord-derived MSCs; MB-MSCs: menstrual blood derived MSCs; LPS: lipopolysaccharide; IV: intra venous; EAE: experimental autoimmune encephalitis; WJ-MSCs: Wharton’s Jelly mesenchymal stem cells.

**Table 2 ijms-23-09274-t002:** Clinical trials about the use of allogeneic MSCs in inflammatory and septic states classified according to indications and trial phases.

Indications	Study phase or type	MSC type	Patients, n	Dose	Results	References
ARDS	Phase I trial	AT-MSCs	Control: 6 Experimental: 6	1 × 10^6^ cells/kg	Safety and feasibility of an AT-MSCs single infusion in treatment of ARDS	Zheng et al. [197]
ARDS	Phase I trial	BM-MSCs	9	1, 5 and 10 × 10^6^ cells/kg	A single infusion of allogeneic BM-MSCs is well tolerated in patients with moderate to severe ARDS	Wilson et al. [194]
ARDS	Phase I trial	UC-MSCs	9	1, 5 and 10 × 10^6^ cells/kg	Safety of a single infusion of UC-MSCs	Yip et al. [198]
H7N9-ARDS	Phase I trial	MB-MSCs	Control: 44 Experimental: 17	1 × 10^6^ cells/kg in 3 or 4 injections	No harmful effects observed	Chen et al. [199]
SARS-CoV-2 ARDS	Phase I trial	UC-MSCs	1 (Case report)	50 × 10^6^ cells/kg × 3 injections	Good tolerance of allogenic UC-MSCs	Liang et al. [195]
SARS-CoV-2 ARDS	Case report	UC-MSCs	Control: 3 Experimental: 7	1 × 10^6^ cells/kg	No adverse effects observed	Leng et al. [196]
ARDS	Phase I	BM-MSCs	Control: 20 Experimental: 40	10 × 10^6^ cells/kg	Safety of MSCs infusion	Matthay et al. [200]
Various critical illness conditions	Meta-analysis	UC-MSCs	93 peer-reviewed full articles or abstracts	various	No long-term adverse effects, tumor formation or cell rejection founded	Can et al. [201]
Septic shock	Phase I	BM-MSCs	Control: 21 Experimental: 9	0.5, 1 and 3 × 10^6^ cells/kg	Infusion of freshly cultured allogenic BM-MSCs up to 3 × 10^6^ cells/kg seems safe	Mcintyre et al. [202]
Severe sepsis	Phase I	UC-MSCs	Control: 15 Historical case-matched: 15	1, 2 and 3 × 10^6^ cells/kg	No infusion-associated serious events or treatment-related adverse events	He et al. [192]
LPS-mediated sepsis (LPS at 2 ng/kg) 1 h after MSC infusion)	Phase I	AT-MSCs	32 (healthy subjects)	0.25, 1 and 4 × 10^6^ cells/kg	IV infusion of AT-MSCs at a dose of 4.10^6^ cells/kg is well tolerated and associated with various procoagulant, pro and anti-inflammatory effects	Perlee et al. [193]
Septic shock	Phase I	BM-MSCs	Control: 21 Historical case-matched group: 9	0.3, 1 and 3 × 10^6^ cells/kg	Safe response characterized by the absence of elevation of plasma-cytokine levels	Schlosser et al. [203]
ARDS (the START study)	Phase IIa	BM-MSCs	Control: 20 Experimental: 40	10 × 10^6^ cells/kg	Significant decrease of Angiopoietin-2, a marker of endothelial dysfunction. No survival improvement	Matthay et al. [200]
Septic shock in severe neutropenic patients (the RUMCESS study)	Phase II	BM-MSCs	Control: 15 Experimental: 15	1 × 10^6^ cells/kg	Good tolerance and safety of MSCs infusion in neutropenic patients. A faster hemodynamic stabilization, vasopressor with- drawal, attenuation of respiratory failure and shortening of the neutropenia duration period	Gennadiy et al. [191]

Abbreviations. MSCs: mesenchymal stem cells; ARDS: acute respiratory distress syndrome; AT-MSCs: adipose tissue-derived MSCs; BM-MSCs: bone marrow-derived MSCs; UC-MSCs: umbilical cord-derived MSCs; MB-MSCs: menstrual blood-derived MSCs; LPS: lipopolysaccharide; IV: intra venous.

## Data Availability

Not applicable.

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
