# Peer review of "Pathophysiology of Sepsis and Genesis of Septic Shock: The Critical Role of Mesenchymal Stem Cells (MSCs)"

_ijms, 2022, doi:10.3390/ijms23169274_

Round 1

Reviewer 1 Report

This is an excellent review article on an important topic. Other than requesting to make minor grammatical, typographical, and stylistic corrections to the language, I have no comments.

Author Response

We would like to thank Reviewer 1 for his/her comments and suggestions. In response to the minor criticisms, we have carried out a careful revision and proofreading of the text of the manuscript in order to improve its grammar, syntax and corrected spelling mistakes. We hope that the changes made will suit the reviewer 1 to satisfaction.

Dr Daniel

Reviewer 2 Report

The review links regulation of innate and adaptive immunity mechanisms and tissue repair with activities of mesenchymal stem cells (MSC) in sepsis. In the review, a popular concept of dysregulated host responses in sepsis is successfully complemented by contribution of MSC accumulated in vicinity of  inflammation areas. The major dignity of the discussed results is the intensive reviewing of recent achievements in studies that employ allogeneic MSC  in sepsis animal models and clinical trials. The most updated clinical trials list for today in this area presented in the review  provides comprehensive  information on perspectives of clinical use of MSC products in acute respiratory distress syndrome that commonly complicates sepsis, and COVID-19. MSC and MSC-derived products are discussed as a challenging paradigm that should impact developing novel therapeutic strategies for sepsis patients. 

Some shortcomings includes  too many uninformative colors of Table 2. The Table 2  may become shorter in width (there is a potential to decrease the width by re-arranging column four ("patients, n" seems enough; "group" may be omitted). Further, two MSC-based clinical trials that include COVID-19  patients have been included in the Table 2. Although in the next part of the review the exosomes are discussed as the next step in MCS-based cell-free therapy,  too limited data  is provided on clinical trials of the COVID-19 that employ exosomes. Recent clinical studies shows that there are a dozen of COVID-19 clinical studies that use MCS exosomes (for example, reviewed by Anand Krishnan et al., 2022). The paper may benefit  from brief discussion of most recent advances in this field directly related to the subject of the review.

Author Response

We would like to thank Reviewer 2 for his/her insightful comments. We have modified table 2 as requested by replacing the title of column 4 with "patients, n" and deleting superfluous text. We have also included, as highlighted in the comments, a paragraph describing the results of latest yet major works related to the used of MSC's exosome vesicles (EV) in COVID-19. Preclinical and clinical data concerning EVs derived from MSCs are presented in dedicated paragraphs (lines 822 to 833 and 834 to 864) as suggested by reviewer 2. We hope these changes will meet full expectations.   Dr Daniel
